# Emergency Department as the Entry Point to Inpatient Care: A Nationwide, Population-Based Study in South Korea, 2016–2018

**DOI:** 10.3390/jcm10081747

**Published:** 2021-04-17

**Authors:** June-sung Kim, Dong Woo Seo, Youn-Jung Kim, Seok In Hong, Hyunggoo Kang, Su Jin Kim, Kap Su Han, Sung Woo Lee, Sungwoo Moon, Won Young Kim

**Affiliations:** 1Department of Emergency Medicine, Asan Medical Center, University of Ulsan College of Medicine, Seoul 05505, Korea; jsmeet09@gmail.com (J.-s.K.); leiseo@gmail.com (D.W.S.); yjkim.em@gmail.com (Y.-J.K.); finefigs@gmail.com (S.I.H.); 2Department of Emergency Medicine, Hanyang University College of Medicine, Seoul 04763, Korea; emer0905@gmail.com; 3Department of Emergency Medicine, Korea University College of Medicine, Seoul 02841, Korea; icarusksj@gmail.com (S.J.K.); hanks96@hanmail.net (K.S.H.); kuedlee@korea.ac.kr (S.W.L.); 4Department of Emergency Medicine, Korea University Ansan Hospital, Ansan 15355, Korea; yg9912@korea.ac.kr

**Keywords:** hospital admission, epidemiology, hospital utilization, emergency department

## Abstract

(1) Background: The emergency department provides lifesaving treatment and has become an entry point to hospital admission. The purpose of our study was to describe the characteristics and outcomes of patients who were admitted through the emergency department to the intensive care unit or general ward. (2) Methods: We performed a retrospective, cross-sectional, descriptive analysis using the National Emergency Department Information System, analyzing patient data including disease category, diagnosis, and mortality from 1 January 2016, to 31 December 2018. (3) Results: During the study period, about 13.6% were admitted through the emergency department. Of these, the overall in-hospital mortality was 4.6%. The frequent disease class for the intensive care unit admissions was the cardiovascular system, and the classes for the general ward admissions were as follows: injury and toxicology, digestive system, and respiratory system. Cardiovascular system-related emergencies were the predominant cause of death among patients admitted to the intensive care unit; however, oncologic complications were the leading cause of death in the general ward. (4) Conclusions: Emergency departments are incrementally utilized as the entry point for hospital admission. Health care providers need to understand emergency department admission epidemiology and prepare for managing patients with certain common diagnoses.

## 1. Introduction

Emergency departments (EDs) have a crucial role in the health care system, acting as hospital admission gatekeepers and a place for the provision of lifesaving treatment. Recent studies have proven that immediate interventions and admission through an ED for various diseases, such as ST-segment elevation myocardial infarction, acute ischemic stroke, and geriatric trauma, could improve outcomes [1,2,3]. The utility of EDs has been increasing and has outpaced the growth of the general population. This has been caused primarily by an increase in the proportion of aging patients with greater rates of chronic diseases [4,5]. Admissions through an ED have also increased simultaneously [6]. Between 2003 and 2009 in the United States of America (USA), hospital admissions originating in the ED increased by 17%, whereas admissions from physicians’ offices and clinics decreased by 10% [7]. Furthermore, another study announced that 70% of hospital admissions were processed through the ED, and there was a linear association between age and the ED admission in the USA [8]. The characteristics of admissions through the ED greatly differ from those of regularly scheduled admissions. Patients requiring emergency admission were typically older, had more comorbidities, and had severe acute illnesses [9]. Moreover, there were potential differences in emergency admitted patients between the general ward (GW) and intensive care unit (ICU) [10,11]. Assessing the epidemiology of patients admitted through an ED is essential to prepare and manage hospital bed resource; however, data on disease classification, specific diagnosis, and mortality in patients admitted from the ED to the hospital are limited. Kwak et al. conducted a retrospective study and described the nation-wide ED utilization pattern by children in Korea [12]. Chen et al. compared the epidemiological characteristics and disease spectrum of patients of two EDs in China [13]. However, these studies did not provide specific diagnoses, classifications, and mortality for both the ICU and GW [12,13].

The primary purpose of this study was to describe common disease classifications and specific diagnoses in patients admitted to the hospital from the ED between 2016 and 2018. We also examined the frequent specific diagnoses in non-survival patients to provide insights for preparing future medical resources.

## 2. Materials and Methods

### 2.1. Study Design

This retrospective, cross-sectional, descriptive, nationwide study was conducted using the National Emergency Department Information System (NEDIS) between 1 January 2016, and 31 December 2018. NEDIS was started in 2003 by the Ministry of Health and Welfare and is managed by the National Emergency Medical Center of Korea. NEDIS covers all clinical and administrative data on patients who visit EDs throughout the country to maintain ED quality and enhance the emergency medical service system [14]. It is regarded as a reliable source of ED data due to the fact that >98% of EDs participate in the system and annual governmental review of the data. This study was approved by the Institutional Review Board of the study facility, and informed consent was waived because of the anonymous nature of the data.

### 2.2. Study Setting and Population

According to the Korean Emergency Medical Care Act, there are three categories of EDs, ranging from large to small regarding the size and role of EDs, i.e., regional emergency centers, local emergency centers, and local emergency institutes. The NEDIS has a variable to differentiate the hospitals according to the levels. We included all admitted patients through EDs of both regional emergency centers and local emergency centers. Patients were excluded if they had visited local emergency institutes because the characteristics of patient population and disease spectrum were totally different from others. Most local emergency institutes in Korea have no ICUs and find it hard to manage patients with critical illnesses. In addition, we excluded patients who were discharged, died, transferred to other hospitals in the ED, or had missing outcome data, such as lost to follow-up. Available variables were extracted, such as demographics, insurance status, route of arrival (direct from the scene or home, transfer from other outpatient clinic or hospital), causes of ED visits (medical or trauma), acuity at the initial triage, and location of admission. According to the Korean Triage and Acuity Scale (KTAS), acuity at the initial triage was rated between 1 and 5, with level 1 indicating the most severely ill patients [15,16]. Length of stay, in line with the previous study, was defined as the interval (in minutes) between patient arrival at the ED and subsequent departure [17].

### 2.3. Measures

We compared the number and rate of admissions by location (i.e., ICU and GW) and collected clinical outcomes on the admitted patients, including discharge, death, and transfer. Others (0.7%) included all unclassified cases discharged without adequate recommendations and patients who ran away. We used the final diagnosis on admission according to the Korean Standard Classification of Disease and Cause of Death (KCD) codes [18]. KCD codes are managed by Statistics Korea, a central organization for statistics under the Ministry of Strategy and Finance, and identical to the International Classification of Disease 10. Furthermore, the diseases and causes of death were classified based on Statistics Korea.

### 2.4. Data Analysis

All statistical analyses were performed with R statistics software, version 3.5.0 (R Foundation for Statistical Computing, Vienna, Austria). Continuous parameters were presented as means and standard deviations (S) and compared using the Student’s t-test or the Mann–Whitney U test. Categorical parameters were expressed as a number and percentage and compared with chi-square tests. *p* values of < 0.05 were considered to be statistically significant.

## 3. Results

A total of 27,483,303 patients visited EDs throughout Korea (9,127,979 vs. 9,089,055 vs. 9,266,269, 2016, 2017, 2018, respectively) during the study period, of these, 21,844,366 (83.7%) were discharged, 145,250 (0.6%) died in the EDs, and 467,592 (1.8%) were transferred to other hospitals (Figure 1). Of the 3,539,945 (13.6%) ED admissions (599,434 (92.3%) for ICU and 2,950,511 (11.3%) for GW), 3,004,984 (86.1%) resulted in discharge, 162,065 (4.6%) died during hospital stay, and 295,706 (8.5%) were transferred to other hospitals after discharge. The total admission number and rates continuously increased from 1,144,407 (12.5%) in 2016 to 1,215,171 (13.1%) in 2018. Furthermore, the absolute number of admissions to the GW continuously increased from 2016 (*n* = 943,737) to 2018 (*n* = 1,021,534) (Figure 2). Meanwhile, the number of admissions to the ICU decreased from 2016 (*n* = 200,607) to 2018 (*n* = 193,637).

### 3.1. Baseline Characteristics of the Study Population

Table 1 shows the baseline characteristics of patients admitted through the ED according to the admission location. Male sex was predominant with a mean age of 53.4 years old. Direct visits (61.1 for ICU vs. 68.9% for GW) and transfers from outpatient departments (2.5 vs. 5.2%) were frequent in the GW group, and transfers from other hospitals (36.4 vs. 25.9%) were common in the ICU group. Medical problems were prevalent in both groups (80.6 vs. 82.7%), and mean ED length of stay was significantly shorter in the ICU group than in the GW group (353.2 vs. 455.0 min). KTAS 1 and 2, called severe illness, were more common in the ICU group (10.0 vs. 0.8% for KTAS 1, 37.5 vs. 10.5% for KTAS 2).

### 3.2. Frequencies of Disease Classes

Table 2 shows the frequencies of disease classes according to admission location. Over half (59.5%) of all cases comprised injury (17.2%), and emergencies related to the digestive system (15.2%), circulatory system (13.4%), and respiratory system (12.7%). The circulatory system (35.3%) was the leading cause of admission in the ICU, followed by injury (16.8%), and respiratory system (10.2%). Injury (17.3%) and digestive system (16.3%) were common in the GW, and a higher proportion of patients were admitted in the GW for neoplasms (9.1%) than in the ICU (4.0%).

### 3.3. Common Specific Diagnoses

Table 3 shows the common specific diagnoses of admitted patients. Pneumonia (0.045%) was the most common cause in the total population, followed by acute appendicitis, cerebral infarct, and acute kidney injury. Hemorrhagic stroke (0.088%) and acute myocardial infarction (0.083%) were predominant in the ICU, and pneumonia (0.043%) and acute appendicitis (0.032%) were frequent in the GW.

### 3.4. Causes of Death Classification

Figure 3 shows the most common causes of death classification. Neoplasms were the leading causes of death in the total (30%) and GW (48%) group. Meanwhile, diseases of the circulatory system had the largest proportion of deaths (32%), followed by respiratory system emergencies (19%), and injury (12%) in the ICU group.

### 3.5. Causes of Death Diagnoses

Figure 4 shows the specific KCD codes for in-hospital deaths. Pneumonia was the most frequent diagnosis in both the ICU (*n* = 9392, 12.3%) and GW (*n* = 11,009, 12.9%). Sepsis (*n* = 4601, 6.1%) was second-most common, followed by hemorrhagic stroke (*n* = 4450, 5.9%), and cardiac arrest (*n =* 3825, 5.0%) in the ICU admission. Meanwhile, incidence of cancers, including lung cancer (*n* = 7279, 8.5%) and hepatocellular carcinoma (*n* = 4759, 5.5%), were more common than that of sepsis (*n* = 2279, 2.6%) in patients admitted to the GW.

## 4. Discussion

Information on common causes of admission through the ED and the cause of death provide insight into a community’s health care system. Our results clearly showed (1) 13.6% (2.3% ICU and 11.3% GW) of patients were admitted through the ED, (2) cardiovascular system-related emergencies comprised more than one-third of the total ICU admission cases, and oncologic diseases were the leading causes of GW admission, (3) around one-twentieth died during admission and pneumonia was the most common diagnosis of death for both ICU and GW, and (4) cardiovascular system-related emergencies, such as acute stroke, cardiac arrest, and myocardial infarct were the predominant cause of death in the ICU; however, oncologic complications were the leading cause of death in the GW, and digestive issues and injuries were relatively less common.

In the 2017 Emergency Department Benchmarking Alliance (EDBA) report, the performance measures survey showed that the average proportion of patients admitted to hospital from the ED was 16.9%, and their length of stay was 303 min in the USA. Our study confirmed that Korea’s ED admission rate is approximately 13%, which increased from 12.5% in 2016 to 13.1% in 2018. This growth was primarily due to the rise in admissions to the GW. Previous epidemiologic studies of ED utilization on the detailed categorization of patients with entire emergency admission records are limited. Recent nationwide analyses in the USA reported emergency care-sensitive conditions and showed sepsis, chronic obstructive pulmonary disease, pneumonia, and heart failure were predominant [19]. In line with previous research, we found that cardiovascular, pulmonary, gastrointestinal, and oncologic issues were the leading causes of urgent admissions. Moreover, pneumonia, cerebral infarction, acute kidney injury, and heart failure were the common attributing diagnoses on admission. These may reflect the geriatric populations with multiple comorbidities as the mainstay of hospital resources. One nationwide descriptive study in Korea with pediatric visits announced that 15.5% of patients were admitted to the ED and fever was the most common symptom [12]. However, they did not provide information on the detailed diagnosis or classification for admission.

We additionally found a common mortality rate and cause of death among admitted patients through the ED. One of the benefits of the NEDIS registry is that death information is included and can be used to calculate nationwide ED mortality. Even though mortality could be impacted by varying illness severity and study population, the rate was quite similar to a previous nationwide study in England (4.8%) [20]. Another retrospective multicenter cohort study showed that the mortality of acutely admitted patients increased up to 10-fold compared with the general population [21]. Our results showed that pneumonia was the most common cause of death in both the ICU and GW. Furthermore, cardiovascular accidents were common in the ICU, and cancer-related problems were frequent reasons for mortality in the GW. For example, patients with pneumonia tend to require more medical resources than other diseases, such as high flow nasal cannula, mechanical ventilator, and negative pressure rooms [22]. Moreover, patients with pneumonia could negatively impact ED overcrowding, which worsened patients’ overall outcomes [22]. Regarding acute ischemic or hemorrhagic stroke, urgent interventions, such as administration of the thrombolytic agent, or hemispheric decompression, with emergent admission through ED to ICU, are known to be the single most important treatment to improve neurologic outcome [23]. Meanwhile, Majzoub et al. reported that early palliative care consultations after ED admission were associated with lower risk for hospital death, and limited palliative care resources needed to be directed to those patients who were admitted through ED [24]. These data could give additional insights for preparation of future administration.

This study has several limitations. First, the NEDIS data were collected from large EDs, therefore, the generalization of the results to small hospitals may not be appropriate. However, the significance of the outcomes could provide detailed information on medical resources. Second, the NEDIS database does not disclose each ED’s clinical data, so appropriate patient management could not be confirmed. Third, we used the first NEDIS diagnosis only for both admission and death analyses. While this approach may have deprived analysis of the complete information, we assumed the initial diagnosis would reflect the full situation.

## 5. Conclusions

In conclusion, ED admissions are increasing. Healthcare providers need to understand the epidemiology of ED admissions and prepare to manage these patients with a specific common diagnosis.

## Figures and Tables

**Figure 1 jcm-10-01747-f001:**
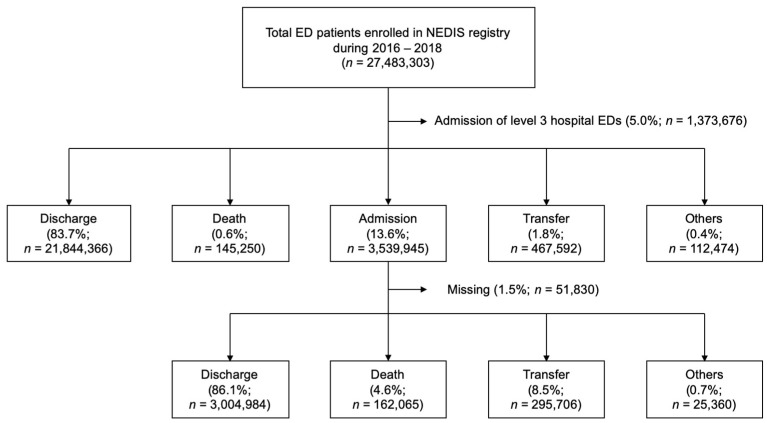
Study flowchart. ED, emergency department; NEDIS, National Emergency Department Information System.

**Figure 2 jcm-10-01747-f002:**
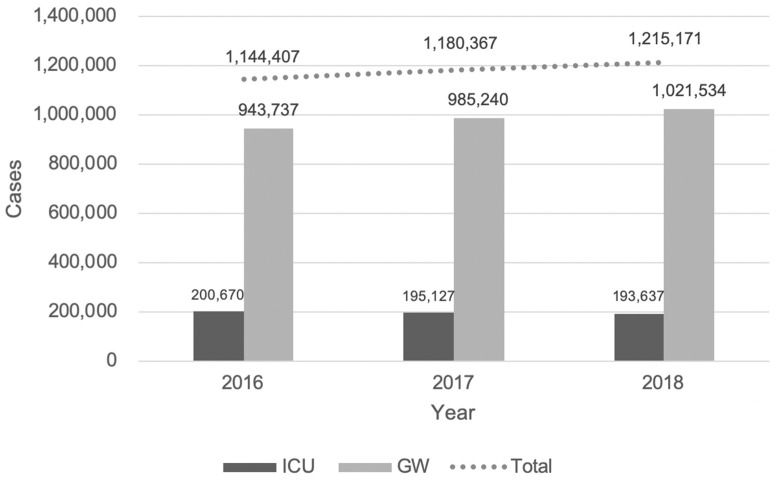
Trends of the total admission through the emergency department during the study period. ED, emergency department; GW, general ward; ICU, intensive care unit.

**Figure 3 jcm-10-01747-f003:**
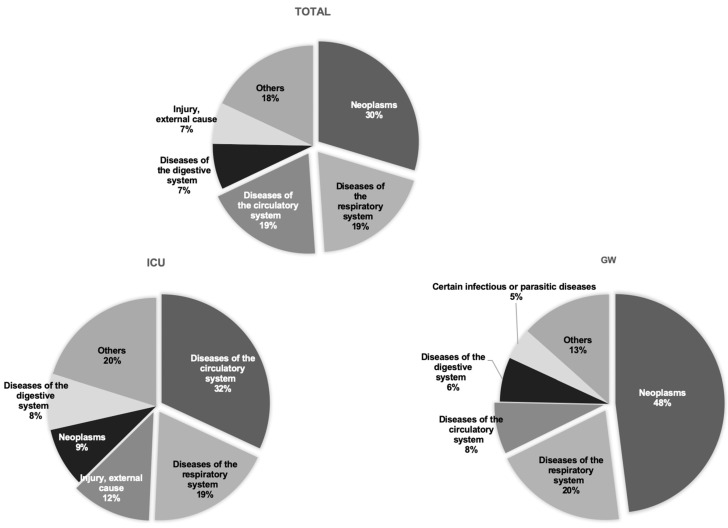
Top 5 disease classifications that lead to death among patients admitted through the emergency department. GW, general ward; ICU, intensive care unit.

**Figure 4 jcm-10-01747-f004:**
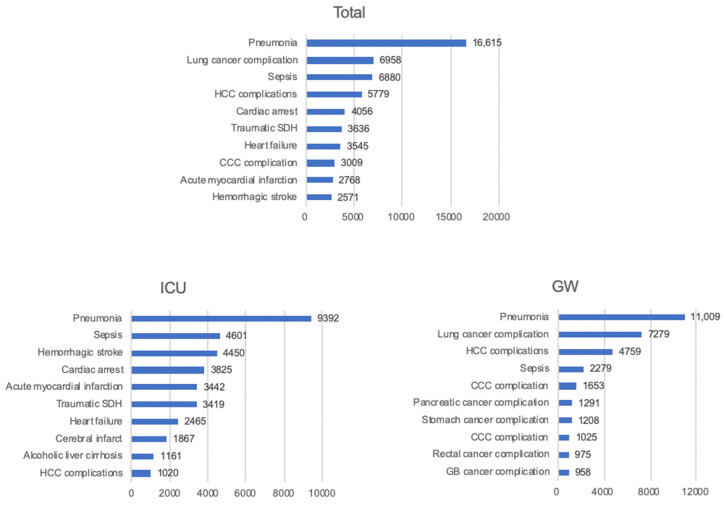
Top 10 diagnosed causes of death among patients admitted through the emergency department. CCC, cholangiocellular carcinoma; HCC, hepatocellular carcinoma; GB, gallbladder; GW, general ward; ICU, intensive care unit; SDH, subdural hematoma.

**Table 1 jcm-10-01747-t001:** Baseline characteristics of the study population.

Variable	Total (*n* = 3,539,945)	ICU (*n* = 589,434)	GW (*n* = 2,950,511)	*p*
Gender				
Male (*n*, %)	1,908,254 (53.9)	350,475 (59.5)	1,557,779 (52.8)	<0.01
Female (*n*, %)	1,631,691 (46.1)	238,959 (40.5)	1,392,732 (47.2)	<0.01
Age (year)	53.4 ± 25.7	62.5 ± 20.2	51.6 ± 26.3	<0.01
<18 (*n*, %)	491,239 (13.9)	25,916 (4.4)	465,323 (15.8)	
18–44 (*n*, %)	638,695 (18.0)	71,253 (12.1)	567,442 (19.2)	
45–64 (*n*, %)	1,013,495 (28.6)	191,046 (32.4)	2,950,511 (27.9)	
≥65 (*n*, %)	1,396,515 (39.5)	301,218 (51.1)	1,095,297 (37.1)	
National insurance (*n*, %)	3,038,589 (85.8)	484,277 (82.2)	2,554,312 (86.6)	<0.01
Route of arrival				<0.01
Directly visit (*n*, %)	2,391,715 (67.6)	359,984 (61.1)	2,031,731 (68.9)	
Referred from OPD (*n*, %)	169,338 (4.8)	14,608 (2.5)	154,730 (5.2)	
From other hospitals (*n*, %)	977,557 (27.6)	214,501 (36.4)	763,056 (25.9)	
Others (*n*, %)	1076 (0.0)	244 (0.0)	832 (0.0)	
Unknown	259 (0.0)	97 (0.0)	162 (0.0)	
Disease type				<0.01
Medical (*n*, %)	2,913,685 (82.3)	474,860 (80.6)	2,438,825 (82.7)	
Trauma (*n*, %)	606,336 (17.1)	97,452 (16.5)	508,884 (17.2)	
Unknown (*n*, %)	19,924 (0.6)	17,122 (2.9)	2802 (0.1)	
EDLOS (min)	438.0 ± 665.1	353.2 ± 741.4	455.0 ± 647.5	<0.01
Level of hospital				<0.01
Regional emergency centers (*n*, %)	1,293,935 (36.6)	252,362 (42.8)	1,041,573 (35.3)	
Local emergency centers (*n*, %)	2,246,010 (63.4)	337,072 (57.2)	1,908,938 (64.7)	
KTAS				<0.01
1 (*n*, %)	83,101 (2.3)	58,811 (10.0)	24,290 (0.8)	
2 (*n*, %)	530,657 (15.0)	221,319 (37.5)	309,338 (10.5)	
3 (*n*, %)	1,846,874 (52.2)	236,511 (40.1)	1,610,363 (54.6)	
4 (*n*, %)	941,739 (26.6)	46,837 (7.9)	894,902 (30.3)	
5 (*n*, %)	115,900 (3.3)	6429 (1.1)	109,471 (3.7)	
Missing (*n*, %)	21,674 (0.6)	19,527 (3.3)	2147 (0.1)	

Data are presented as *n* (%) or mean with standard deviation. ICU, intensive care unit; GW, general ward; OPD, outpatient department; EDLOS, emergency department length of stay; KTAS, Korean Triage Acuity Score.

**Table 2 jcm-10-01747-t002:** Frequencies of disease classes according to admission type.

Rank	Total (*n* = 3,539,945)		ICU (*n* = 589,434)		GW (*n* = 2,950,511)	
	Code	Frequency (%)	Code	Frequency (%)	Code	Frequency (%)
1	Injury	608,555 (17.2)	Circulatory system	207,850 (35.3)	Injury	509,450 (17.3)
2	Digestive system	539,058 (15.2)	Injury	99,105 (16.8)	Digestive system	481,367 (16.3)
3	Circulatory system	472,996 (13.4)	Respiratory system	59,857 (10.2)	Respiratory system	389,908 (13.2)
4	Respiratory system	449,765 (12.7)	Digestive system	57,691 (9.8)	Neoplasms	268,486 (9.1)
5	Neoplasms	291,911 (8.2)	Not classified	33,385 (5.7)	Circulatory system	265,146 (9.0)
6	Genitourinary system	246,785 (7.0)	Genitourinary system	27,721 (4.7)	Infectious disease	221,594 (7.5)
7	Infectious disease	243,932 (6.9)	Neoplasms	23,425 (4.0)	Genitourinary system	219,064 (7.4)
8	Not classified	225,347 (6.4)	Infectious disease	22,338 (3.8)	Not classified	191,962 (6.5)
9	Nervous system	95,396 (2.7)	Nervous system	17,118 (2.9)	Skin disorders	78,314 (2.7)
10	Musculoskeletal system	81,880 (2.3)	Metabolic diseases	16,067 (2.7)	Nervous system	78,278 (2.7)
11	Metabolic diseases	76,389 (2.2)	Contact health services	5976 (1.0)	Metabolic diseases	60,322 (2.0)
12	Pregnancy	42,813 (1.2)	Mental disorders	5051 (0.9)	Pregnancy	40,064 (1.4)
13	Ear disease	32,286 (0.9)	Musculoskeletal system	3566 (0.6)	Ear disease	31,657 (1.1)
14	Mental disorders	31,028 (0.9)	Pregnancy	2749 (0.5)	Diseases of the skin	28,155 (1.0)
15	Skin disorders	29,394 (0.8)	Blood disorders	2042 (0.4)	Mental disorders	25,977 (0.9)
16	Blood disorders	25,806 (0.7)	Perinatal disorders	1720 (0.3)	Blood disorders	23,764 (0.8)
17	Contact health services	18,692 (0.5)	Developmental anomalies	1503 (0.3)	Contact health services	16,972 (0.6)
18	Visual system disease	11,191 (0.3)	Skin disorders	1239 (0.2)	Visual system disease	10,945 (0.4)
19	Perinatal disorders	8375 (0.2)	Ear disease	629 (0.1)	Developmental anomalies	6355 (0.2)
20	Developmental anomalies	7858 (0.2)	Visual system disease	246 (0.0)	Perinatal disorders	2399 (0.1)
21	External causes	343 (0.0)	External causes	132 (0.0)	External causes	211 (0.0)
22	Special purposes	145 (0.0)	Special purposes	24 (0.0)	Special purposes	121 (0.0)

Data are presented as *n* (%). ICU, intensive care unit; GW, general ward.

**Table 3 jcm-10-01747-t003:** Top 20 common diagnoses of admitted patients.

Rank	Total *(n* = 3,539,945)		ICU (*n* = 589,434)		GW (*n* = 2,950,511)	
	Code	Frequency (%)	Code	Frequency (%)	Code	Frequency (%)
1	Pneumonia	158,554 (4.5)	Hemorrhagic stroke	51,982 (8.8)	Pneumonia	127,374 (4.3)
2	Acute appendicitis	95,558 (2.7)	AMI	49,156 (8.3)	Acute appendicitis	93,978 (3.2)
3	Cerebral infarct	92,835 (2.6)	Pneumonia	31,180 (5.3)	Rectal cancer complication	81,405 (2.8)
4	AKI	86,441 (2.4)	Cerebral infarct	28,475 (4.8)	Cerebral infarct	69,191 (2.3)
5	Rectal cancer complications	84,716 (2.4)	Sepsis	12,448 (2.1)	AKI	65,760 (2.2)
6	Urinary tract infection	44,178 (1.2)	Heart failure	10,477 (1.8)	Urinary tract infection	37,558 (1.3)
7	AMI	43,346 (1.2)	AKI	9236 (1.6)	Femur neck fracture	34,465 (1.2)
8	Seizure	37,413 (1.1)	Ischemic heart disease	8091 (1.4)	Enterocolitis	27,266 (0.9)
9	Femur neck fracture	35,461 (1.0)	Cardiac arrest	7831 (1.3)	HCC complication	21,195 (0.7)
10	Hemorrhagic stroke	33,500 (0.9)	Urinary tract infection	6620 (1.1)	Concussion	21,000 (0.7)
11	Enterocolitis	28,729 (0.8)	CKD	5127 (0.9)	Acute cholecystitis	20,888 (0.7)
12	HCC complications	24,339 (0.7)	Gastrointestinal hemorrhage	4890 (0.8)	Fever, unspecified	20,368 (0.7)
13	Heart failure	24,051 (0.7)	Ischemic heart disease	4568 (0.8)	Acute pancreatitis	18,956 (0.6)
14	CKD	23,140 (0.7)	Gastric ulcer	3552 (0.6)	Acute cholangitis	18,529 (0.6)
15	Concussion	22,247 (0.6)	Pulmonary embolism	3519 (0.6)	Neck sprain	18,403 (0.6)
16	Acute cholecystitis	21,880 (0.6)	Rectal cancer terminal complication	3311 (0.6)	CKD	18,013 (0.6)
17	Fever, unspecified	21,255 (0.6)	Seizure	3221 (0.5)	Seizure	17,904 (0.6)
18	Acute cholangitis	20,110 (0.6)	HCC complication	3144 (0.5)	Dizziness	16,801 (0.6)
19	Acute pancreatitis	19,992 (0.6)	Alcoholic liver cirrhosis	2563 (0.4)	Heart failure	16,272 (0.6)
20	Sepsis	19,628 (0.6)	Pulmonary edema	2173 (0.4)	Lung cancer complications	15,201 (0.5)

Data are presented as *n* (%). ICU, intensive care unit; GW, general ward; AMI, acute myocardial infarction; AKI, acute kidney injury; HCC, hepatocellular carcinoma; CKD, chronic kidney disease.

## Data Availability

The data presented in this study are available on request from the corresponding author.

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
