# Peer review of "Emergency Department as the Entry Point to Inpatient Care: A Nationwide, Population-Based Study in South Korea, 2016–2018"

_jcm, 2021, doi:10.3390/jcm10081747_

Round 1
Reviewer 1 Report
The manuscript aims to describe the characteristics and outcomes of patients who were admitted through emergency department in order to understand epidemiology and prepare for managing patients with certain common diagnoses. The manuscript thus addresses a timely and highly relevant topic in the field of emergency medicine.
At the end of the discussion, the authors comment on ways to proceed according to the results obtained, for example, with the diagnosis of pneumonia and possible measures to be taken. Missing the description of some possible measures for example in cancer and cardiovascular system diagnoses that are the majority groups.
Line 122, clarify in sentence the percentage value 61.1 vs 68.9% what is referring to, I mean ICU group comparing GW group. This would make reading the results easier to understand.
Table 1. The values that appear in the table 1 (first row) are equivalent to the Total Admission in the ED from the years 2016, 2017 and 2018. However If I add the values shown in figure 2 for those years, they do not equal the total of admissions that appears in table 1, and the same happens with the ICU and GW groups. This causes a bit of confusion. On the other hand, the total admission ED that shows table 1 (N = 3,539,945), does coincide with the value that appears in figure 1. Check the numbers.
Table 1. Include in the first row along with the N, the symbol (%). Correct the values present in the table including comma between numbers.
Table 2. Correct the values present in the table including comma between numbers.
Table 2. Normalize the columns of the table and the letter so that the codes are written in the same way in the columns total, ICU or GW, Ex: Circulatory system. And align the values with the percentages on the same line for the three subgroups
Line 143. Correction Common specific diangnoses: diagnoses
Table 3. Correct the values present in the table including comma between numbers.Incluir decimals. Adjust the size of the table in the text because it cannot be seen whole
Line 205. Could you include the rate of mortality in a previous nationwide study in England?
Line 212, It is necessary to include the citation of the study to which this sentence refers
References section:
Include the digital object identifier (DOI) for all references where available.
In the main text, reference numbers should be placed in square brackets [ ] and placed before the punctuation; for example [1], [1–3] 253 or [1,3].
Author Response
The manuscript aims to describe the characteristics and outcomes of patients who were admitted through emergency department in order to understand epidemiology and prepare for managing patients with certain common diagnoses. The manuscript thus addresses a timely and highly relevant topic in the field of emergency medicine.
At the end of the discussion, the authors comment on ways to proceed according to the results obtained, for example, with the diagnosis of pneumonia and possible measures to be taken. Missing the description of some possible measures for example in cancer and cardiovascular system diagnoses that are the majority groups.
Response> Thanks for generous comments. We totally agreed with the reviewer’s opinion and added sentences regarding patients with cancer and cardiovascular system diagnoses as follow.
“Our results showed that pneumonia was the most common cause of death in both the ICU and GW. Furthermore, cardiovascular accidents were common in the ICU, and cancer-related problems were frequent reasons for mortality in the GW. For example, patients with pneumonia tend to require more medical resources than other diseases, such as high flow nasal cannula, mechanical ventilator, and negative pressure rooms [22]. Moreover, patients with pneumonia could negatively impact ED overcrowding, which worsened patients’ overall outcomes [22]. Regarding acute ischemic or hemorrhagic stroke, urgent interventions, such as administration of the thrombolytic agent, or hemispheric decompression, with emergent admission through ED to ICU, are known to be the single most important treatment to improve neurologic outcome [23]. Meanwhile, Majzoub et al. assisted that early palliative care consultations after ED admission were associated with lower risk for hospital death, and limited palliative care resources needed to be directed to patients those who admitted through ED [24]. These data could give additional insights for preparation of future administration.” (Line 683-690)
Line 122, clarify in sentence the percentage value 61.1 vs 68.9% what is referring to, I mean ICU group comparing GW group. This would make reading the results easier to understand.
Response> We added and reordered the phrases for clarifying the meaning of the sentences.
“Direct visits (61.1 for ICU vs. 68.9% for GW) and transfers from outpatient departments (2.5 vs. 5.2%) were frequent in the GW group, and transfer from other hospitals (36.4 vs. 25.9%) was common in the ICU group.” (Line 168-170)
Table 1. The values that appear in the table 1 (first row) are equivalent to the Total Admission in the ED from the years 2016, 2017 and 2018. However If I add the values shown in figure 2 for those years, they do not equal the total of admissions that appears in table 1, and the same happens with the ICU and GW groups. This causes a bit of confusion. On the other hand, the total admission ED that shows table 1 (N = 3,539,945), does coincide with the value that appears in figure 1. Check the numbers.
Response> We checked and corrected the error in figure 2.
Table 1. Include in the first row along with the N, the symbol (%). Correct the values present in the table including comma between numbers.
Response> We included the symbol (%) in the first row and inserted comma between numbers in the table 1.
Table 2. Correct the values present in the table including comma between numbers.
Response> We inserted comma between numbers in table 2.
Table 2. Normalize the columns of the table and the letter so that the codes are written in the same way in the columns total, ICU or GW, Ex: Circulatory system. And align the values with the percentages on the same line for the three subgroups
Response> We normalized the columns of the table 2 as the reviewer’s mention.
Line 143. Correction Common specific diangnoses: diagnoses
Response> We corrected typo.
Diangnoses -> diagnoses
Table 3. Correct the values present in the table including comma between numbers.Incluir decimals. Adjust the size of the table in the text because it cannot be seen whole line 205.
Response> We inserted comma between numbers and adjusted the size of the table 3.
Could you include the rate of mortality in a previous nationwide study in England?
Response> We included the rate of mortality in a previous nationwide study in England.
“Even though mortality could be impacted by varying illness severity and study population, the rate was quite similar to a previous nationwide study in England (4.8%)” (Line 672-674)
Line 212, It is necessary to include the citation of the study to which this sentence refers
References section:
Response> We added the reference as the reviewer’s mention.
“Moreover, patients with pneumonia could negatively impact ED overcrowding, which worsened patients’ overall outcomes [22].” (Line 681-683)
Include the digital object identifier (DOI) for all references where available.
Response> We added the available DOIs of all references.
In the main text, reference numbers should be placed in square brackets [ ] and placed before the punctuation; for example [1], [1–3] 253 or [1,3].
Response> We corrected the reference numbers following format of the journal in the main text.

Reviewer 2 Report
Abstract
The abstract contains several precise data, in numbers AND percentage, which does not help the reader, neither convince them to read the paper.
The topic 4 (conclusions) should be enhanced in order to evidence the reasons why a reader should read the paper.
Introduction
Lines. (33-34)
The references 1-3 are not enough discussed to be related with the statement about ED and their role.
Ls. (38-40) Unnecessary double use of “United States”. “United States” should be replaced by “United States of America”.
Ls. (40-41) “In 2017, 17% of ED visits resulted in admission, and 70% of hospital admissions were processed through the ED.” The statement should be enhanced. Where? Why? How?
Ls. (47-50) “however, data on disease classification, specific diagnosis, and mortality in patients admitted from the ED to the hospital are limited. Moreover, previous studies have reported on small populations, and did not provide specific diagnoses and causes of admission for both the ICU and GW 12,13.” Authors should explain why they refer 12 & 13. These papers are no other papers about the subject? The references 12-13 are not enough discussed to be related to the statement.
Ls. (71-75) “Patients were excluded if they had visited local emergency institutes”. Authors should explain how they account patients from local emergency institutes, and why and how they exclude patients from this third category of ED.
Ls. (102-108) “Total admission number and rates continuously increased from 1,144,407 (12.5%) in 2016 to 1,215,171 (13.1%) in 2018.” These numbers are not presented in Figure 2. Authors should enhance the explanation about these data.
Ls.(122-127) “Direct visits (61.1 vs. 68.9%) and transfers from outpatient departments (5.2% vs. 2.5%)...”
The use of % symbol should be accurate.
The first one compares (ICU vs. GW) and the following ones compare the opposite (GW vs. ICU). Authors must normalise the data avoiding to confuse the reader.
Later, authors compare Total vs GW. Authors should explain the comparison change.
L (140) “Table 2. Frequencies of disease classes according to admission type”
must be placed on the same page as the Table 2.
Table 3 is cutted and it is not totally visible.
Figure 3 refers to “neoplasms” and in the text (154) “Malignancy-related problems” is used. Authors should elucidate the reader about the use of these two expressions.
Authors should specify in the text, near each data (%) the corresponding letter of the chart.
Ls (163-168) Statements seem to be inaccurate. Authors should be more precise detailing the data in the charts of Figure 4, identifying the letter of the corresponding chart.
L (177 - ) Statements seem to be inaccurate. Authors should be more precise detailing the data in Table 2, identifying the letter of the corresponding chart.
“one-twentieth died” -> “around one-twentieth died” / “near one-twentieth died”
Author Response
Abstract
The abstract contains several precise data, in numbers AND percentage, which does not help the reader, neither convince them to read the paper.
Response> Thanks for great suggestion. We deleted several precise numbers and percentage as the reviewer’s recommendations.
“Abstract: (1) Background: The emergency department provides lifesaving treatment and has become an entry point to hospital admission. The purpose of our study was to describe the characteristics and outcomes of patients who were admitted through emergency department to the intensive care unit or general ward. (2) Methods: We performed a retrospective, cross-sectional, descriptive analysis using the National Emergency Department Information System, analyzing patient data including disease category, diagnosis, and mortality from January 1, 2016, to December 31, 2018. (3) Results: During study period, about 13.6% were admitted through the emergency department. Of these, overall in-hospital mortality was 4.6%. The frequent disease classes for intensive care unit admission was cardiovascular system, and for general ward admission was as follows: injury and toxicology, digestive system, respiratory system. Cardiovascular system-related emergencies were the predominant cause of death among patients admitted to the intensive care unit; however, oncologic complications were the leading cause of death in the general ward. (4) Conclusions: Emergency departments are incrementally utilized as the entry point for hospital admission. Health care providers need to understand emergency department admission epidemiology and prepare for managing patients with certain common diagnoses.”
The topic 4 (conclusions) should be enhanced in order to evidence the reasons why a reader should read the paper.
Response> We added a sentence for highlighting our findings.
“Conclusions: Emergency departments are incrementally utilized as the entry point for hospital admission. Health care providers need to understand emergency department admission epidemiology and prepare for managing patients with certain common diagnoses.”
Introduction
Lines. (33-34)
The references 1-3 are not enough discussed to be related with the statement about ED and their role.
Response> Thanks for great suggestion. We added more sentence to support the statement about ED with related references [1-3].
“Recent studies have proved that immediate interventions and admission through ED for various diseases, such as ST-segment elevation myocardial infarction, acute ischemic stroke, and geriatric trauma, could improve outcomes [1-3].” (Line 34-36)
Ls. (38-40) Unnecessary double use of “United States”. “United States” should be replaced by “United States of America”.
Response> We deleted unnecessary double use of “United States”, and replaced the word.
“Between 2003 and 2009 in the United States of America, hospital admissions originating in the ED increased by 17%, whereas admissions from physicians' offices and clinics decreased by 10% [7].” (Line 40-57)
Ls. (40-41) “In 2017, 17% of ED visits resulted in admission, and 70% of hospital admissions were processed through the ED.” The statement should be enhanced. Where? Why? How?
Response> We totally agreed with the reviewer’s opinion and changed the sentence for enhancing the meaning of the statement as below.
“Furthermore, another study announced that 70% of hospital admissions were processed through the ED, and there was a linear association between age and the ED admission in the USA [8].” (Line 57-59)
Ls. (47-50) “however, data on disease classification, specific diagnosis, and mortality in patients admitted from the ED to the hospital are limited. Moreover, previous studies have reported on small populations, and did not provide specific diagnoses and causes of admission for both the ICU and GW 12,13.” Authors should explain why they refer 12 & 13. These papers are no other papers about the subject? The references 12-13 are not enough discussed to be related to the statement.
Response> We totally agreed the reviewer’s comment that the references [12,13] were not enough discussed to be related to the statement. We additionally mentioned about the references [12,13] to compare our study and elucidate the purpose of writing this article.
“Kwak et al. conducted retrospective study to describe the nation-wide ED utilization pattern by children in Korea [12]. Chen et al. compared the epidemiological characteristics and disease spectrum of patients of two EDs in China [13]. However, these studies did not provide specific diagnoses, classifications, and mortality for both the ICU and GW [12,13].” (Line 66-71)
Ls. (71-75) “Patients were excluded if they had visited local emergency institutes”. Authors should explain how they account patients from local emergency institutes, and why and how they exclude patients from this third category of ED.
Response> The National Emergency Department Information System (NEDIS) includes a variable for discriminating the level of hospitals. We selected the data of both regional emergency centers and local emergency centers and excluded that of local emergency institutes. The characteristics of patient population and disease spectrum of local emergency institutes are totally different from others because most of institutes have no ICUs and hard to manage patients with critical illnesses. We
"The NEDIS have a variable to discriminate the hospitals according to the levels. We included all admitted patients through EDs of both regional emergency centers and local emergency centers. Patients were excluded if they had visited local emergency institutes because the characteristics of patient population and disease spectrum were totally different from others. Most local emergency institutes in Korea have no ICUs and hard to manage patients with critical illnesses.” (Line 92-98)
Ls. (102-108) “Total admission number and rates continuously increased from 1,144,407 (12.5%) in 2016 to 1,215,171 (13.1%) in 2018.” These numbers are not presented in Figure 2. Authors should enhance the explanation about these data.
Response> The original numbers in figure 2 were incorrected, and we revised the figure.
Ls.(122-127) “Direct visits (61.1 vs. 68.9%) and transfers from outpatient departments (5.2% vs. 2.5%)...”
The use of % symbol should be accurate.
The first one compares (ICU vs. GW) and the following ones compare the opposite (GW vs. ICU). Authors must normalise the data avoiding to confuse the reader.
Later, authors compare Total vs GW. Authors should explain the comparison change.
Response> Thanks for the generous comments. We normalized the orders of the variables for avoiding to confuse the reader. We also deleted inaccurate the use of % symbol.
“Direct visits (61.1 for ICU vs. 68.9% for GW) and transfers from outpatient departments (2.5 vs. 5.2%) were frequent in the GW group, and transfer from other hospitals (36.4 vs. 25.9%) was common in the ICU group. Medical problems were prevalent in both groups (80.6 vs. 82.7%), and Mean ED length of stay was significantly shorter in the ICU than in the GW (353.2 vs. 455.0 minutes). KTAS 1 and 2, called severe illness, were more common in the ICU group (10.0 vs. 0.8% for KTAS 1; 37.5 vs. 10.5% for KTAS 2).” (Line 168-174)
L (140) “Table 2. Frequencies of disease classes according to admission type”
must be placed on the same page as the Table 2.
Response> We placed legend with table 2 on the same page as the reviewer’s recommendation.
Table 3 is cutted and it is not totally visible.
Response> We changed the size of the table 3.
Figure 3 refers to “neoplasms” and in the text (154) “Malignancy-related problems” is used. Authors should elucidate the reader about the use of these two expressions.
Response> We accepted the reviewer’s suggestion and changed the word as below.
“Neoplasms were the leading causes of death in the total (30%) and GW (48%) group.” (Line 448-449)
Authors should specify in the text, near each data (%) the corresponding letter of the chart.
Ls (163-168) Statements seem to be inaccurate. Authors should be more precise detailing the data in the charts of Figure 4, identifying the letter of the corresponding chart.
Response> We rewrite the data regarding to the charts of figure 4 as the reviewer’s comments.
“Pneumonia was the most frequent diagnosis in both the ICU (n = 9,392, 12.3%) and GW (n = 11,009, 12.9%). Sepsis (n = 4,601, 6.1%) was second-most common, followed by hemorrhagic stroke (n = 4,450, 5.9%), and cardiac arrest (n = 3,825, 5.0%) in the ICU admission. Meanwhile, incidence of cancers, including lung cancer (n = 7,279, 8.5%), and hepatocellular carcinoma (n = 4,759, 5.5%), were more common than that of sepsis (n = 2,279, 2.6%) in patients admitted to the GW.” (Line 608-614)
L (177 - ) Statements seem to be inaccurate. Authors should be more precise detailing the data in Table 2, identifying the letter of the corresponding chart.
Response> We agreed the reviewer’s opinion and changed the data and percentage according to the table 2 as follow.
“Table 2 shows the frequencies of disease classes according to admission location. Over half (59.5%) of all cases comprised injury (17.2%), and emergencies related to the digestive system (15.2%), circulatory system (13.4%), and respiratory system (12.7%). The circulatory system (35.3%) was the leading cause of admission in the ICU, followed by injury (16.8%), and respiratory system (10.2%). Injury (17.3%) and digestive system (16.3%) were common in the GW, and a higher proportion of patients were admitted in the GW for neoplasms (9.1%) than in the ICU (4.0%).” (Line 193-199)
“one-twentieth died” -> “around one-twentieth died” / “near one-twentieth died”
Response> We inserted the word as recommendation.
“around one-twentieth died” (Line 637)
